# Embrittlement Due to Excess Heat Input into Friction Stir Processed 7075 Alloy

**DOI:** 10.3390/ma12020227

**Published:** 2019-01-10

**Authors:** Ming-Hsiang Ku, Fei-Yi Hung, Truan-Sheng Lui

**Affiliations:** Department of Materials Science and Engineering, National Cheng Kung University, Tainan 701, Taiwan; n58051049@mail.ncku.edu.tw (M.-H.K.); fyhung@mail.ncku.edu.tw (F.-Y.H.)

**Keywords:** 7075 aluminum alloy, friction stir process (FSP), tool rotational speed, tensile ductility, ductile-to-brittle transition (DBT)

## Abstract

The grain size of high strength 7075 hot-rolled aluminum plates was refined by a friction stir process (FSP) to improve their mechanical properties. The results of the tensile ductility tests, which were conducted at various tool rotational speeds, in the friction stir zone indicate significant tensile ductility loss, which even resulted in a ductile-to-brittle transition (DBT). DBT depends on the tool rotational speed. Our 1450 rpm specimens showed large data fluctuation in the tensile ductility and the location of the fracture controlled the formation of friction stir induced bands (FSIB). The crack initiation site located at FSIB was due to the tool rotational speed (1670 rpm). A higher heat-input causes the formation of FSIB, which is accompanied with micro-voids. This contributes significantly to tensile cracking within the stir zone after the application of the aging treatment. This investigation aimed to determine the dominant factor causing tensile ductility loss at the stir zone, which is the major restriction preventing further applications.

## 1. Introduction

Aluminum alloys 7075, which are based on the Al-Zn-Mg-Cu system, are widely used for transportation facilities [1,2,3] due to their good combination of strength, hardness and fracture toughness. However, it is difficult to join 7075 aluminum alloy by traditional welding due to the defects (such as shrinkage voids and dendritic structures) that occur in the fusion zone during the solidification [4].

Fusion welding is widely used for metals but results in poor weldability. The mechanical attributes produced by friction stir processing (FSP) have attracted considerable attention in the recent years. FSP is a solid-state joining process that does not produce the defects of traditional fusion processes. FSP [5,6,7] employs a rotating tool that moves along the substrate to generate heat, which results in severe plastic deformation. The microstructural observations of the friction stir zone (SZ) described the presence of uniformly fine, dynamically recrystallized grains, which was thus given the name of continuous dynamical recrystallization (CDRX) mechanism [8,9,10]. After performing FSP technology, the microstructural features (including grain size and the morphology of the intermetallic particles) of a specimen are greatly improved in terms of mechanical properties [11,12,13,14,15].

Besides, this can be applied to a wider area by multipass friction stir processing [16,17,18,19], which is used as a forming (aid) process. Numerous studies have investigated the metallurgical factors and the related mechanical properties of friction stir processed materials. Chegeni et al. [20] investigated the effects of friction stir welding (FSW) on the microstructural and mechanical properties of 7075 aluminum rolled plates in addition to post welding heat treatment (PWHT) in the semi-solid region. Al-Fadhalah et al. [21] examined the effect of artificial aging on the microstructure, texture and hardness homogeneity of 6082 aluminum alloy via FSP. Barbini et al. [22] also examined the correlation between heat generation, microstructure and mechanical properties of dissimilar AA2024/AA7050 FSW joints. However, only a few studies have focused on the ductile-to-brittle transition phenomenon. This issue is especially apparent when the materials suffer from a higher frictional heat input.

An increase in the rotational speed is always needed to accelerate the production rate. Many researches have investigated the effect of rotational speed. Yang et al. [23] reported that rotational speed affects the width and the hardness of material flow band in AA2024-T351 and AA2524-T351 aluminum alloys. Rajakumar et al. [24] indicated that a higher tool rotational speed easily produced micro-voids on the upper surface of the stir zone of 7075 aluminum alloy. Furthermore, the presence of voids has an effect on the tensile properties. Elangovan et al. [25] indicated that the rotational speed and the pin profile affect the tensile properties during FSP because of the formation of voids. Our previous report [26] indicated that the rotational speed affects the ductility because of non-uniform element distribution. This issue is especially apparent when the materials suffer from excess heat-input. However, there is a lack of detail related to the correlations between the material flow, the voids and precipitates as well as their effect on tensile properties and fracture behaviors.

However, an increase in rotational speed is often accompanied with many risks, such as the deterioration of material ductility. As for the excess heat-input, Frigarrd et al. [27] suggested an equation describing the input under an identical moving speed as q = 4/3π^2^uPNR^3^, where q is the heat, u is the friction coefficient, P is the pressure, N is the rotational speed and R is the radius of the shoulder.

From the examination of the crack initiating site during tensile deformation within the stir zone (SZ), it has been determined that embrittlement is often due to two common mechanisms: (1) the presence of heterogeneously nucleated particles and (2) the occurrence of strain and intense shear bands. Both factors are governed by the combined effect of micro-void formation with assembled intermetallic particles and friction stir induced bands (FSIB). One way to improve tensile ductility under an excess heat-input condition of FSP involves the development of a control cooling method [15,28]. However, with this array of research having focused on the emerged phase due to excess heat-input, minimal effort has been directed toward understanding the deformation kinetics of fine compound particles in 7075 aluminum alloy matrix by observing the nature of the crack initiation.

In this investigation, we aimed to obtain a comprehensive understanding of the effects of aging and friction stir rotational speed up to a hyper-rotational condition of 1670 rpm. This will allow us to study the tensile deformation and fracture development pertaining to the occurrence of friction stir induced bands (FSIB) and micro-voids. In addition, we increased the rotational speed to ensure that there was excess heat-input for examining the microstructural features during the ductile-to-brittle transition (DBT).

## 2. Materials and Methods

In this study, a commercial hot rolled 7075 aluminum plate with a thickness of 5 mm was used. All hot rolled 7075 Al alloy specimens underwent solution heat treatment (T4-753 K, 1 h) and T6 artificial aging treatment (393 K, 24 h) to yield 7075-T4/T6 aluminum alloy as the base metal (BM-T4/T6). The chemical composition of 7075 Al alloy was measured by an optical emission spectrometer, the results of which are shown in Table 1.

A schematic illustration of our FSP is shown in Figure 1. The FSP processing direction was parallel to the rolling direction. The employed tool had a pin diameter of 6 mm, a shoulder diameter of 18 mm and a pin length of 3.3 mm. The chosen rotational speed ranged from 440 to 1670 rpm while the moving speed of the tool was fixed at 0.58 mm s^−1^ with a 1.5° tool angle and a downward pushing force of 38.7 MPa. The FSP specimen at 1230 rpm was designated as 1230 and the other abbreviations for FSP specimens were assigned likewise.

The FSP specimens received three PWHTs: natural aging (313 K, 100 h), solution heat treatment (T4-753K 1 h and subsequent water quenching) and T4/T6 heat treatment (T4-753 K, 1 h and T6-393 K, 24 h, respectively). The 1230 rpm FSP specimens after these treatments were designated as “1230-NA”, “1230-W” and “1230-T4/T6”, respectively. Abbreviations for FSP specimen names were assigned likewise. The microstructural features of FSP specimens were examined by optical microscope (OM). The morphology and chemical composition of the precipitate and the intermetallic particles were examined by transmission electron microscopy (TEM, Tecnai F20 G2, EFI, Hillsboro, OR, USA), scanning electron microscope, (SEM, HITACHI SU-5000, HITACHI, Tokyo, Japan) energy dispersive spectrometer (EDS, EDAX, Singapore, Singapore) and electron probe X-ray microanalyzer (EPMA, JEOL JXA-8900R, JEOL, Tokyo, Japan).

The sampling position and a schematic illustration of the tensile specimens are shown in Figure 2. The cross-section of the tensile specimens had dimensions of 4 mm × 2 mm and the gauge length was 10 mm. The strain rate was fixed at 1.67 × 10^−3^ s^−1^ at room temperature.

## 3. Results and Discussion

### 3.1. Critical Condition of Ductile-to-Brittle Transition (DBT) Due to Excess Heat-Input

Figure 3 shows the tensile properties obtained with a constant moving speed at room temperature as a function of rotational speed. Figure 3a,b indicates little difference in the yield strength (YS) and ultimate tensile strength (UTS) when increasing frictional rotational speed from 440 rpm to 1230 rpm. It is likely that the actual ductile-to-brittle transition of the 7075 alloy appears at 1450 rpm because of a large data fluctuation in the tensile ductility as shown in Figure 3c,d. Hence, we have chosen samples that become brittle in the hyper-rotational range and examined whether they exhibit the DBT phenomenon. Figure 3 reveals a noticeable difference in both tensile strength and ductility when the rotational speed increases from 1450 rpm to 1670 rpm.

Examination should focus on the rotational speed range (1230–1670 rpm) in order to find a definite indication of DBT. After the aging treatment, the specimens processed by FSP at 1670 rpm are already in the brittle domain but those at 1230 rpm recover ductility thereafter (Figure 4). The 1450 rpm specimens show large data fluctuation. It should be noted that the specimens freeze under the solution heat treatment condition (W specimen) without showing any ductility loss (include 1670 rpm). A more detailed examination and explanation of the microstructure of FSP specimens will be provided later.

### 3.2. Emergence of FSIB and Micro-Voids on the Susceptibility of DBT

A typical through-thickness microstructure displays grains created by fine equiaxial dynamic recrystallization, as shown in Figure 5, which demonstrates heterogeneity with depth. According to the degree of thermal, mechanical and plastic deformation as well as the heat-input during friction stir processing, the average grain size at the SZ region only varies slightly (from 5.2 μm to 5.0 μm), even the specimens that were T4 heat treated and under an excess heat-input region at 1670 rpm. The fine grain size was maintained with almost no significant coarsening (about 6.0 μm).

As determined by the EPMA examination shown in Figure 6, there was a distinct microstructural difference between 1230 specimen and 1670 specimen, which was namely a friction stir induced band (FSIB) that emerged due to excess heat-input. Figure 6b,c shows the difference of the element distribution between 1230 specimen and 1670 specimen. The element distribution of the 1230 specimen is more uniform than the 1670 specimen. This suggests that the excess heat-input will cause inhomogeneity.

Figure 4 demonstrated significant data fluctuation (1450 rpm) in tensile ductility (among 18 specimens). This suggests that the FSIB formation is the most critical contributor to the variation of fracture behavior.

According to a previous report [24], a higher tool rotational speed resulted in higher heat generation and easily produced micro-voids on the upper surface in the stir zone in 7075 aluminum alloy. Figure 7 reveals a considerable number of intermetallic particles and micro-voids that were aligned with the traces of the metal flow during FSP. These particles and voids resulted in cracking during an earlier tensile deformation stage. Experimental evidence is shown in Figure 8. All FSP specimens undergo segregation of intermetallic to align with the tracing direction due to friction stir rotation, which creates an oval-shaped morphology in most cases. However, in the hyper-rotational speed specimens (1670 rpm), a large number of assembled particles and micro-voids were found to gather in the vicinity of FSIB. SEM/EDS was employed for the identification of compounds, as shown in Figure 8b,c and Table 2. This determined that the primary compounds are Al_7_Cu_2_Fe and Al-Cu-Fe. Figure 7 shows two discontinuous particle lines across each other (circle).

Regarding the effect of aged hardening, the crack initiation site can be identified at a line crossing location (circle) during tensile deformation. Figure 9 reveals the fracture surface of 1670-NA specimens. Some micro-voids and many fine dimples were observed on the fracture surfaces (Figure 9a). There are many precipitates around the fine dimples (Figure 9b). TEM/EDS was employed for the identification of intermetallic particles. As shown in Figure 10 and Figure 11, there are many precipitates located in the matrix and grain boundary and their main components are Al, Zn, Mg and Cu, which resulted in Al_2_CuMg formation. The precipitates have been well-established [9] as η Mg(Zn, Cu, Al)_2_ (rod type). In addition, there is a considerable number of intermetallic particles containing Cr that are located around η MgZn_2_ in the matrix and grain boundary.

From the results of tensile ductility in Figure 4, the segregation of intermetallic particles was not affected by the tensile ductility loss. However, the precipitates have a significant impact on tensile ductility loss. According to a previous report [26], the precipitate was found to affect the difference in hardness between the FSIB and matrix. In this study, we also found that the interaction between those precipitates and micro-voids during the tensile test had an impact on the tensile ductility loss. In other words, the ductility of the lower rotational speed specimens was better than that of the higher rotational speed specimens, even if the intermetallic particles were clustered in the matrix. However, increasing the rotational speed resulted in excess heat-input and easily generated the FSIB and micro-voids. After aging, due to the interactions between precipitates, the FSIB and the micro-voids resulted in the embrittlement during tensile deformation.

The precipitates will increase the deformation resistance of the matrix adjacent to the FSIB and micro-voids. Figure 12 shows the microstructure of the crack near the fracture subsurface of 1670-NA specimen after the tensile test. The results indicated that the crack was produced near onion rings. The hardness near the crack was analyzed by the HV test, as shown in Figure 12a. The results showed that the hardness of A side was higher than that of B side. The FSIB near the crack was also observed by EPMA analyzation, as shown in Figure 12b.

The cracking induced by FSIB due to excess heat-input (1670 specimen) is also associated with another factor that is triggered by strain localization in the vicinity of FSIB areas, which is indicated in Figure 7. The experimental results from the above examinations not only enrich the knowledge pool of FSP but also provide solid clues related to the main crack formation, growth, and coalescence of micro-voids, which is finally attributed to the tensile ductility loss. A schematic diagram of tensile ductility loss due to excess heat-input of friction stir process is shown in Figure 13.

## 4. Conclusions

We observed that the average grain size did not change with an increase of the rotational speed to 1670 rpm and after through solutionizing temperature at 753 K.The dependence of the ductile-to-brittle transition (DBT) phenomenon on rotational speed was confirmed. An increase in the rotation speed to 1450 rpm resulted in the brittle fracture pattern being found in a large proportion of the sample while the obtained tensile ductility showed large data fluctuation.The intermetallic particles were frequently found in the microstructure observations of the specimens that were friction stirred at various rotational speeds. The intermetallic particles were segregated along the string metal flow. However, in the hyper-rotational speed region, the FSIBs were accompanied by the discontinuous intermetallic particles and micro-voids were found due to the excess heat-input.This study clarified that the main reason behind the embrittlement during tensile deformation was the interaction between the precipitates, the FSIB and the micro-voids after aging with the hardening treatment.

## Figures and Tables

**Figure 1 materials-12-00227-f001:**
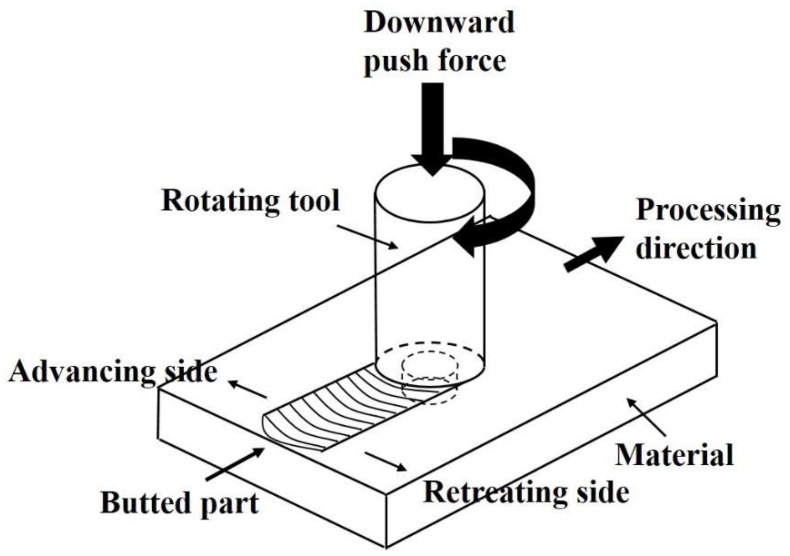
Schematic illustration of friction stir process (FSP).

**Figure 2 materials-12-00227-f002:**
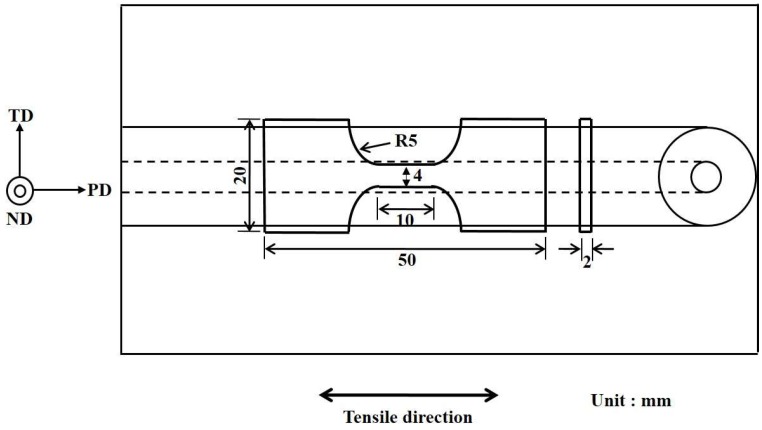
Schematic illustration of the tensile specimen (FSP). (processing direction: PD; normal direction: ND; transverse direction: TD).

**Figure 3 materials-12-00227-f003:**
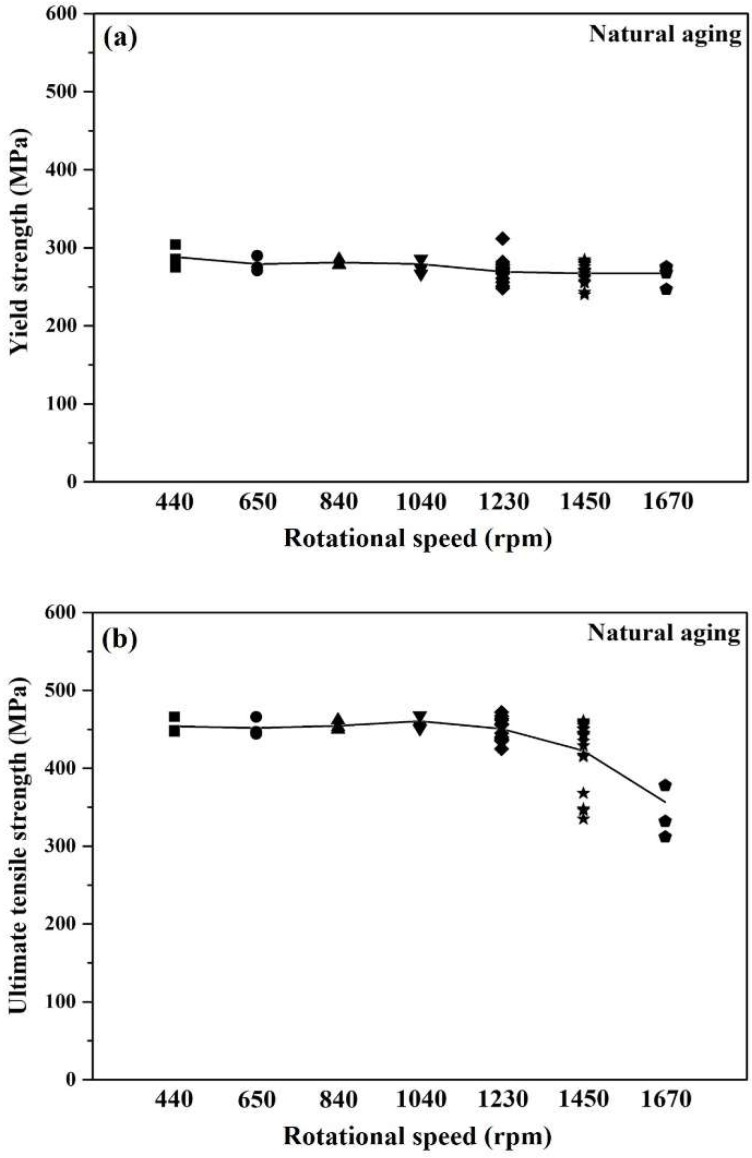
Tensile test of FSP specimens at various rotational speeds after natural aging: (**a**) yield strength (YS); (**b**) ultimate tensile strength (UTS); (**c**) uniform elongation (UE); and (**d**) total elongation (TE).

**Figure 4 materials-12-00227-f004:**
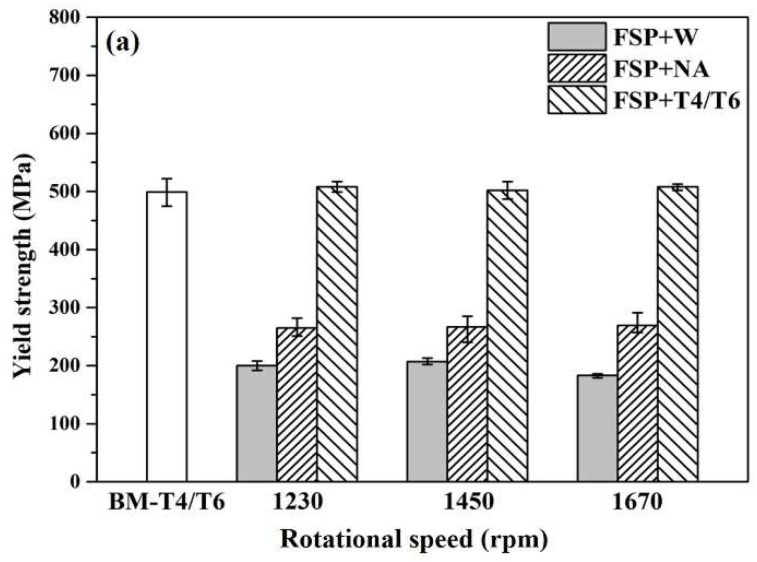
Room-temperature tensile properties of FSP specimens at various tool rotational speeds after PWHTs: (**a**) yield strength (YS); (**b**) ultimate tensile strength (UTS); (**c**) uniform elongation (UE); and (**d**) total elongation (TE). BM-T4/T6 is the raw material used in this FSP experiment.

**Figure 5 materials-12-00227-f005:**
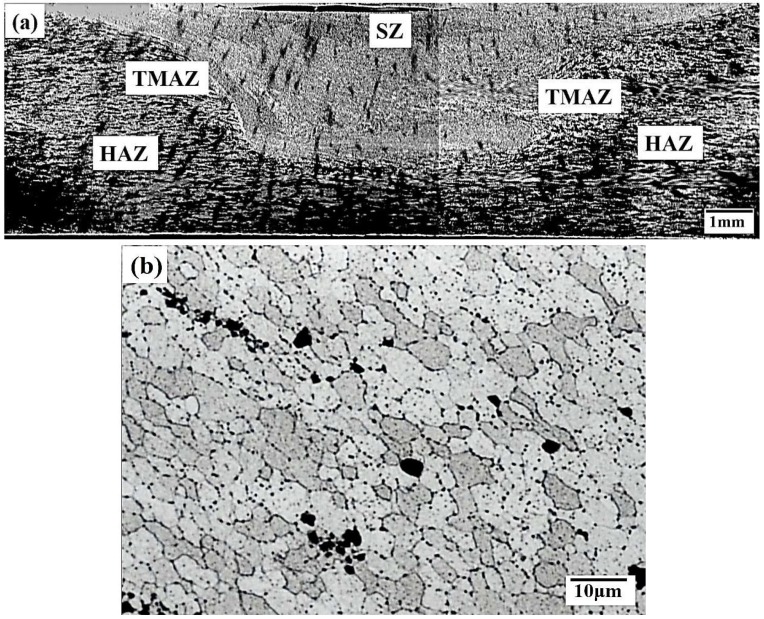
Cross-section of the PD plane for specimens processed at 1450 rpm: (**a**) microstructural zone formation; and (**b**) average grain size of 5.1 μm in the SZ. (SZ: stir zone; TMAZ: thermo mechanically affected zone; HAZ: heat affected zone)

**Figure 6 materials-12-00227-f006:**
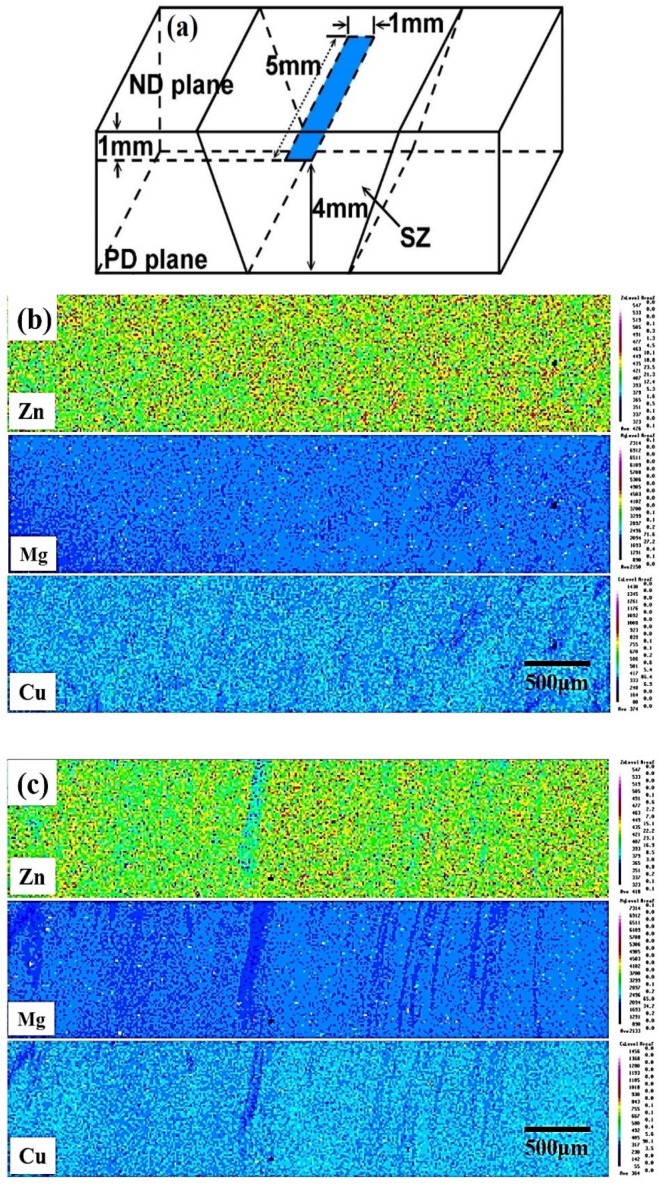
EPMA of the FSP specimens: (**a**) location of the sample; (**b**) 1230 rpm without FSIB structure; and (**c**) 1670 rpm with FSIB structure.

**Figure 7 materials-12-00227-f007:**
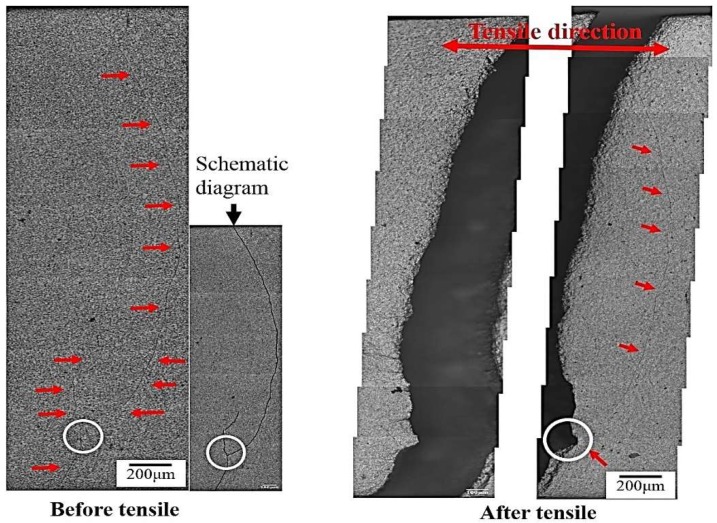
Microstructure of tensile deformation of the 1670-NA specimen (discontinuous arrangement assembled by intermetallic particles and micro-voids (arrow) and two discontinuous intermetallic particle lines crossing each other (circle)).

**Figure 8 materials-12-00227-f008:**
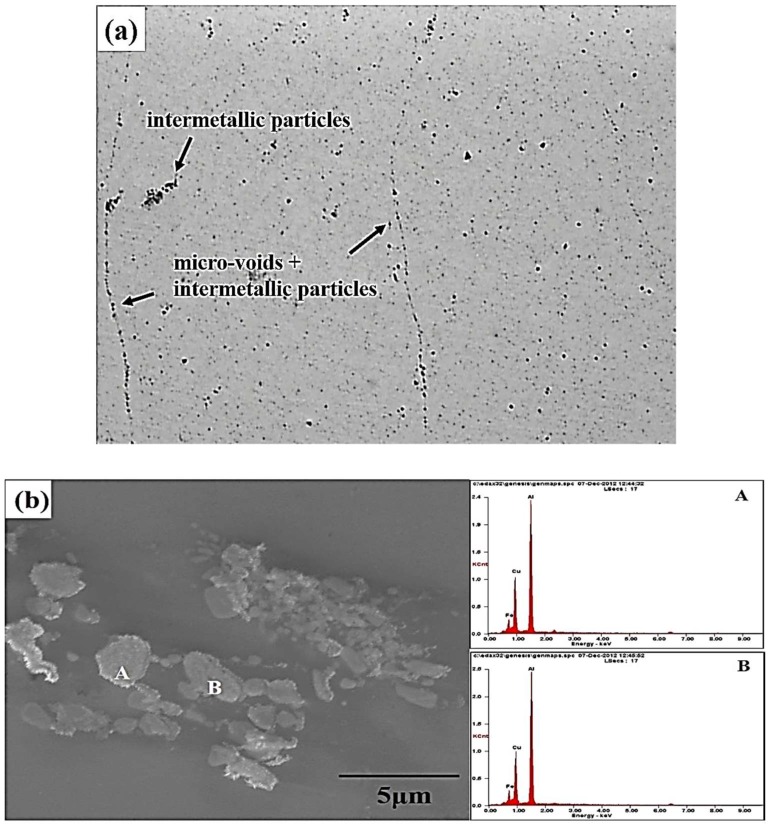
Microstructural morphology and analysis of the intermetallic particles and micro-voids: (**a**) microstructures; and (**b**,**c**) micro-voids and intermetallic particles at different locations.

**Figure 9 materials-12-00227-f009:**
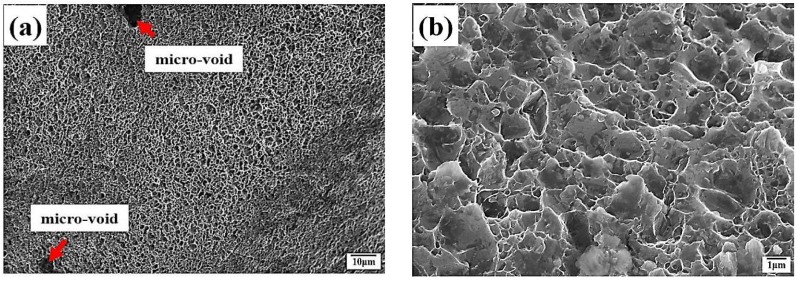
The morphology of fracture surface in the 1670-NA specimens: (**a**) low magnification; and (**b**) high magnification.

**Figure 10 materials-12-00227-f010:**
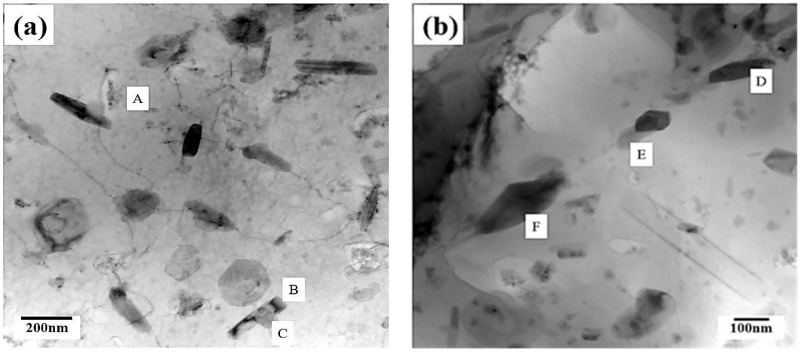
The precipitate analysis of the 1670-NA specimen by TEM: (**a**) matrix; and (**b**) grain boundary.

**Figure 11 materials-12-00227-f011:**
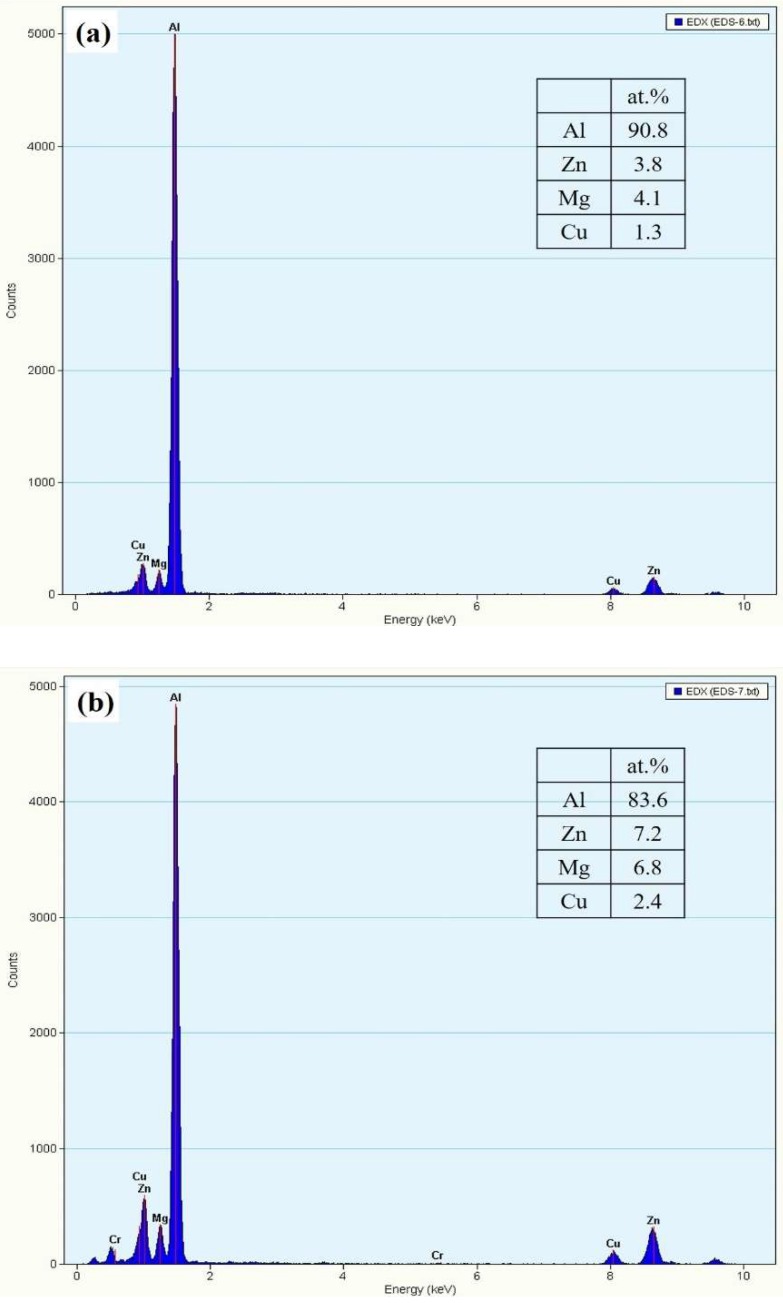
Energy dispersive spectrometer (EDS) for examined specimens, which is shown in Figure 10: (**a**) A point; (**b**) B point (**c**) C point; (**d**) D point; (**e**) E point; and (**f**) F point.

**Figure 12 materials-12-00227-f012:**
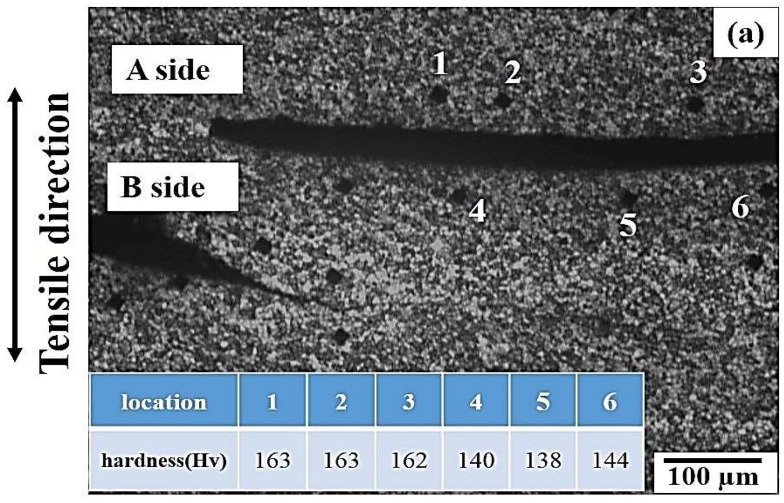
The fracture subsurface of the 1670-NA specimen near the crack: (**a**) the measurement of the hardness (HV) and (**b**) EPMA.

**Figure 13 materials-12-00227-f013:**
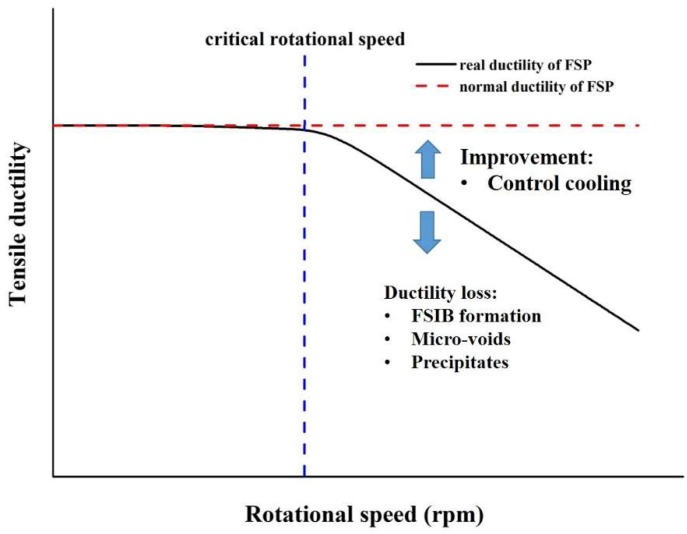
A schematic diagram of tensile ductility improvement during hyper-rotational applications of FSP.

**Table 1 materials-12-00227-t001:** Chemical composition of the 7075-T4/T6 Aluminum alloy (mass. %).

Zn	Mg	Cu	Cr	Fe	Ti	Si	Mn	Al
5.67	2.44	1.71	0.26	0.16	0.02	0.05	0.01	Bal

**Table 2 materials-12-00227-t002:** Engergy dispersive spectrometer (EDS) for examined specimens, which is shown in Figure 8b,c.

No.	Al	Zn	Mg	Cu	Fe
A	72.0	0	0	18.7	9.3
B	73.4	0	0	17.3	9.3
C	57.9	0	0	34.3	7.8
D	94.9	2.8	2.3	0	0

Unit: at.%

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
