# Peer review of "Embrittlement Due to Excess Heat Input into Friction Stir Processed 7075 Alloy"

_materials, 2019, doi:10.3390/ma12020227_

Round 1
Reviewer 1 Report
The paper has to be revised carefully. The following major comments should be considered:
What is the most significant and new points in your work compared to other previous studies. Should be well indicated. The main objective of this study should be clear indicating the new vision and findings of this study applied on such alloys investigated.
The role of second phase particles and precipitates formed are not discussed well in this manuscript.
TEM analysis is not clear in this manuscript to indicate the authors point of view. Should be considered. Electron diffraction patterns should be attached to TEM Figure 10. This figure is not clear enough and should be well indicated.
The microstructure characerization part has to be rewritten carefully and well discussed indicating the the role of critical heat input.
The conclusions should be revised carefully to indicate the most significant and impact points in this work and its main objective .
Author Response
Thank you for your email-letter. We appreciate the reviewers’ valuable comments regarding our paper. Please find enclosed our detailed response and revised paper.
Response to the Reviewer’s Comments
What is the most significant and new points in your work compared to other previous studies. Should be well indicated. The main objective of this study should be clear indicating the new vision and findings of this study applied on such alloys investigated.
Our response:
We greatly appreciated your mention. We amended the context of introduction, including adding the description of the relationship between heat-input, material flow and voids during FSP in the literatures [23]; [24]; [25] (lines 47-53). In addition, we also added more details about the main objective of this study in lines 38-45, lines 55-56, and lines 70-74. All details are as following.
lines 47-53
Yang et al. [23] reports that the effect of rotation speed affects the width and the hardness of material flow band in AA2024-T351 and AA2524-T351 aluminum alloy. Rajakumar et al. [24] indicated that a higher tool rotational speed easily produced micro voids on the upper surface of the stir zone of 7075 aluminum alloy. Furthermore, the existence of voids affect the tensile properties. Elangovan et al. [25] indicated that the rotational speed and the pin profile affect the tensile properties during FSP because of the formation of voids.
lines 38-45
Chegeni et. al. [20] investigated the effects of friction stir welded on the microstructural and mechanical properties of 7075 aluminum rolled plate along with post welding heat treatment in the semi-solid region. Al-Fadhalah et al. [21] examined the effect of artificial aging on the microstructure, texture and hardness homogeneity of 6082 aluminum alloy via FSP. Barbini et al. [22] also examined the correlation of heat generation, microstructure and mechanical properties of dissimilar AA2024/AA7050 FSW joints. However, little investigation focused on the ductile-to-brittle transition phenomenon. This issue is especially apparent when the materials suffer from higher frictional heat input.
lines 55-56
But, more details pertaining the relationships among the material flow, the voids and precipitates on tensile properties and fracture behaviors were not investigated.
lines 70-74
In this investigated the comprehensive understanding for the effects of aging and the effect of friction stir rotation up to a hyper-rotation condition of 1670 rpm in order to study the local tensile deformation and fracture development pertaining to the occurrence of friction stir induced bands (FSIB), micro-voids. In addition, we expand the rotational speed to excess heat-input range to examine the microstructural feature about the occurrence of the ductile-to-brittle transition (DBT).
The role of intermetallic particles and precipitates formed are not discussed well in this manuscript.
Our response:
We greatly appreciated your mention. We amended the discussion of the intermetallic particles and the precipitates, including lines 163-167, and lines 205-214 in this manuscript.
.
lines 163-167
Experiment evidences were shown in Figure 8, all FSP specimens have intermetallic particles segregation to align with the tracing direction due to friction stir rotation, with oval-shaped morphology in most cases. However, in the hyper-rotation speed specimens (1670 rpm), lots of assembled particles and micro-voids gathering could be recognized in the vicinity of FSIB.
lines 205-214
From the results of tensile ductility in Figure 4, the intermetallic particles segregation was not affected on the tensile ductility loss. But the precipitates is an important factor for tensile ductility loss. According to previous report [26], the precipitate was produced to affect the difference of hardness between the FSIB and matrix. In this study, we also found that the interaction between those precipitates and micro-voids during tensile test made an impact on the tensile ductility loss. In other words, the ductility of the lower rotational speed specimens was better than that of the higher rotational speed specimens, even if the intermetallic particles clustered in the matrix. However, as increasing the rotational speed result in excess heat-input and easily generated the FSIB and micro-voids. After aging, owing to the interaction among the precipitates, the FSIB and the micro-voids resulted in the embrittlement during tensile deformation.
TEM analysis is not clear in this manuscript to indicate the authors’ point of view. Should be considered. Electron diffraction patterns should be attached to TEM Figure 10. This figure is not clear enough and should be well indicated.
Our response:
We greatly appreciated your mention. In this investigation, the composition of the intermetallic particles and their distribution have been identified by TEM/EDS. In addition, we also have confirmed that the precipitates play an important role on the tensile ductility loss.
The microstructure characterization part has to be rewritten carefully and well discussed indicating the role of critical heat input.
Our response:
We greatly appreciated your mention. We amended the part of results and discussion pertaining microstructure characterization, including lines 159-161, lines 163-167, and lines 205-214 in this manuscript.
lines 159-161
According to the report [24], they indicated that a higher tool rotational speed resulted in higher heat generation and easily produced micro voids on the upper surface in the stir zone in 7075 aluminum alloy.
lines 163-167
Experiment evidences were shown in Figure 8, all FSP specimens have intermetallic particles segregation to align with the tracing direction due to friction stir rotation, with oval-shaped morphology in most cases. However, in the hyper-rotation speed specimens (1670 rpm), lots of assembled particles and micro-voids gathering could be recognized in the vicinity of FSIB.
lines 205-214
From the results of tensile ductility in Figure 4, the intermetallic particles segregation was not affected on the tensile ductility loss. But the precipitates is an important factor for tensile ductility loss. According to previous report [26], the precipitate was produced to affect the difference of hardness between the FSIB and matrix. In this study, we also found that the interaction between those precipitates and micro-voids during tensile test made an impact on the tensile ductility loss. In other words, the ductility of the lower rotational speed specimens was better than that of the higher rotational speed specimens, even if the intermetallic particles clustered in the matrix. However, as increasing the rotational speed result in excess heat-input and easily generated the FSIB and micro-voids. After aging, owing to the interaction among the precipitates, the FSIB and the micro-voids resulted in the embrittlement during tensile deformation.
The conclusions should be revised carefully to indicate the most significant and impact points in this work and its main objective.
Our response:
We greatly appreciated your mention. We amended the more details for the conclusions, including line 235-248 in this manuscript.
lines 235-248
The average grain size observed barely changed as increasing rotational speed up to 1670 rpm and after through solutionizing temperature at 753 K.
Ductile-to-brittle transition (DBT) phenomenon depends on rotational speed was confirmed. As raising up the rotational speed up to 1450 rpm, a great part of sample with brittle fracture pattern could be recognized, obtained tensile ductility showed a very large fluctuation of data.
The intermetallic particles commonly show in the microstructure observations of the specimen friction stirred by various rotational speed. The intermetallic particles segregated along string metal flow. However, in the hyper rotational speed region, the FSIBs accompany with the discontinuous intermetallic particles and micro-voids could be observed due to the excess heat-input.
This study clarified that the main reason resulting in the embrittlement during tensile deformation is attributable to the interaction among the precipitates, the FSIB and the micro-voids after aging hardening treatment.

Reviewer 2 Report
There is already wide research done by now in the Embrittlement due to excess heat-input and implemented in many industries worldwide. Authors need to consider this.
Author Response
Thank you for your email-letter. We appreciate the reviewers’ valuable comments regarding our paper. Please find enclosed our detailed response and revised paper.
Response to the Reviewer’s Comments
There is already wide research done by now in the embrittlement due to excess heat-input and implemented in many industries worldwide. Authors need to consider this.
Our response:
We greatly appreciated your mention. There is already wide research done by now in the embrittlement due to excess heat-input. However, most of researches only investigated the relationship between the microstructural features and heat input. Hence, in this study, we investigated the comprehensive understanding for the effects of aging and the effect of friction stir rotation up to a hyper-rotation condition of 1670 rpm in order to study the local tensile deformation and fracture development pertaining to the occurrence of friction stir induced bands (FSIB), micro-voids. In addition, we expand the rotational speed to excess heat-input range to examine the microstructural feature about the occurrence of the ductile-to-brittle transition (DBT). In addition, we amended more details on the introduction, results and discussion and conclusions, including lines 47-53, lines 38-45, lines 55-56, lines 159-161, lines 163-167, lines 205-214, and lines 235-248.

Reviewer 3 Report
This manuscript simply describes the effect of the rotational speed on the microstructure and mechanical properties, and suggests some mechanism such as FSIB, microvoids, and ppts for ductility loss, but it is not clearly shown if this mechanism is attributed to the excess heat input.
The author should show more clear relaltionship between excess heat input and birttle behavior of FSPed alloys.
Author Response
Thank you for your email-letter. We appreciate the reviewers’ valuable comments regarding our paper. Please find enclosed our detailed response and revised paper.
Response to the Reviewer’s Comments
This manuscript simply describes the effect of the rotational speed on the microstructure and mechanical properties, and suggests some mechanism such as FSIB, microvoids, and ppts for ductility loss, but it is not clearly shown if this mechanism is attributed to the excess heat input. The author should show more clear relationship between excess heat input and brittle behavior of FSPed alloys.
Our response:
We greatly appreciated your mention. We amend more details the relationship between excess heat input and brittle behavior of FSPed alloy, including lines 159-161, lines 163-167, and lines 205-214 in this manuscript.
lines 159-161
According to the report [24], they indicated that a higher tool rotational speed resulted in higher heat generation and easily produced micro voids on the upper surface in the stir zone in 7075 aluminum alloy.
lines 163-167
Experiment evidences were shown in Figure 8, all FSP specimens have intermetallic particles segregation to align with the tracing direction due to friction stir rotation, with oval-shaped morphology in most cases. However, in the hyper-rotation speed specimens (1670 rpm), lots of assembled particles and micro-voids gathering could be recognized in the vicinity of FSIB.
lines 205-214
From the results of tensile ductility in Figure 4, the intermetallic particles segregation was not affected on the tensile ductility loss. But the precipitates is an important factor for tensile ductility loss. According to previous report [26], the precipitate was produced to affect the difference of hardness between the FSIB and matrix. In this study, we also found that the interaction between those precipitates and micro-voids during tensile test made an impact on the tensile ductility loss. In other words, the ductility of the lower rotational speed specimens was better than that of the higher rotational speed specimens, even if the intermetallic particles clustered in the matrix. However, as increasing the rotational speed result in excess heat-input and easily generated the FSIB and micro-voids. After aging, owing to the interaction among the precipitates, the FSIB and the micro-voids resulted in the embrittlement during tensile deformation.

Round 2
Reviewer 1 Report
The authors did most of required modifications and corrections. The manuscript can be accepted in its revised form.
Reviewer 2 Report
please do minor final revision
Reviewer 3 Report
The reviewer appreciates authors' effort for completeness of the manuscript.